# Chemotaxonomy of Southeast Asian *Peperomia* (Piperaceae) Using High-Performance Thin-Layer Chromatography Colour Scale Fingerprint Imaging and Gas Chromatography–Mass Spectrometry

**DOI:** 10.3390/plants13192751

**Published:** 2024-09-30

**Authors:** Yutthana Banchong, Theerachart Leepasert, Pakawat Jarupund, Trevor R. Hodkinson, Fabio Boylan, Chalermpol Suwanphakdee

**Affiliations:** 1Department of Botany, Faculty of Science, Kasetsart University, Bangkok 10900, Thailand; yutthana.b@ku.th; 2Department of Chemistry, Faculty of Science, Kasetsart University, Bangkok 10900, Thailand; fscitcl@ku.ac.th; 3National Biobank of Thailand (NBT), National Science and Technology Development Agency (NSTDA), Pathum Thani 12120, Thailand; pakawat.jar@ncr.nstda.or.th; 4Botany, School of Natural Sciences, Trinity College Dublin, D02 PN40 Dublin 2, Ireland; hodkinst@tcd.ie; 5School of Pharmacy and Pharmaceutical Sciences, Trinity Biomedical Sciences Institute, Trinity College Dublin, D02 PN40 Dublin 2, Ireland; fabio.boylan@tcd.ie; 6The Trinity Centre for Natural Product Research (NatPro), Trinity College Dublin, D02 PN40 Dublin 2, Ireland

**Keywords:** chemometric, chromatogram, image analysis, tropical plants, HPTLC, GC-MS

## Abstract

The morphological characters of Southeast Asia’s indigenous *Peperomia* species are very similar, especially in their flower structures. The flowers are simple, hermaphrodite and lack a perianth. Therefore, many species are hard to distinguish using morphological characters alone. Here, we apply chemometric data for species identification and classification, gathered using multiwavelength detection combined with the colour scale High-Performance Thin-Layer Chromatography (HPTLC) fingerprinting procedure and chemical compounds determined by Gas Chromatography–Mass Spectrometry (GC-MS). Fourteen taxa were investigated using hexane, ethyl acetate and ethanol solvent extractions. Principal component analysis (PCA) and hierarchical cluster analysis (HCA) were used with the colour scale fingerprints to classify the *Peperomia* species. The PCA and HCA using the chromatogram profile from hexane divided the taxa into six groups compared to the profile from ethyl acetate and ethanol, which each detected seven groups. The chromatogram from the combined dataset of all three solvents can differentiate all the species. The GC-MS data detected a total of 40 compounds from the hexane extract, and these differed among *Peperomia* species. This approach based on HPTLC fingerprinting and GC-MS analysis can therefore be used as a tool for authentication and identification studies of *Peperomia* species.

## 1. Introduction

*Peperomia* is the second-largest genus in the family Piperaceae, containing 1500–1700 species with a pantropical distribution [1]. *Peperomia* are annual or perennial herbs. They can be epiphytic, epilithic or terrestrial and are often found in high and moist forests. Most of the species have similar morphological characteristics, and species identification is difficult because the reproductive organs are very small, often caducous and sometimes lacking. The vegetative organs are highly variable among and within species [2]. The taxonomic delimitation of *Peperomia* is further complicated because some characters differ when examined from fresh samples or from dried herbarium specimens of the same taxon. Reproductive organs in particular are often lost when specimens are dried [3]. 

Taxonomic work in Southeast Asia has nevertheless revealed a significant species richness [4,5,6,7,8,9]. Previous studies reported five species in Indo-China [4], seven species in the Malay Peninsula [5] and seven species in Java [6]. Recently, 17 native species were reported in Thailand [7]. In traditional medicine, *Peperomia* species have been used to treat asthma, coughs, ulcers, conjunctivitis, inflammation and high cholesterol and have functioned as diuretics, analgesics and antibiotics [8]. In the current global trend, it has been reported that medicinal plants are widely used as substitutes for new therapeutic drugs because many medicinal plants contain active compounds that aid in treatment while having no side effects [9,10,11]. Notably, several species in the genus *Peperomia* have been reported to possess therapeutic properties [8]. Chemotaxonomic studies of *Peperomia* are therefore of significant interest in pharmacognosy. They also support the taxonomic identification of vegetative material that is in fragmented or poor condition or in cases where reproductive organs are absent. Various analytical techniques have been developed for medicinal plant identification and quality control, including marker-based, multicomponent-based and pattern recognition or fingerprinting approaches [12,13,14,15,16]. Fingerprint profiling is increasingly recognised as an effective method to describe the complex composition of medicinal plant extracts, facilitating the evaluation of identity, authenticity and consistency of herbal products by demonstrating similarities and differences between samples [17]. Organisations such as the World Health Organisation (WHO) [18], the American Food and Drug Administration (FDA) [19] and the European Medicines Agency (EMA) [20] have endorsed fingerprint analysis as a strategy for identifying and characterising herbal drugs and assessing batch-to-batch consistency of plant-based medicines. Fingerprint analysis has emerged as a very useful technique to assess the quality of herbal drug materials and formulations for establishing standardised herbal products.

Chromatographic techniques like high-performance thin-layer chromatography (HPTLC) and gas chromatography–mass spectrometry (GC-MS are widely employed to develop characteristic fingerprint profiles [21,22]. These analytical tools are particularly valuable in phytochemical applications. Here, we assess the composition of plant extracts using HPTLC (including multiwavelength imaging) with three solvents (ethanol, ethyl acetate and hexane) and GC-MS on the hexane extracts alone. Hierarchical cluster analysis (HCA) and principal component analysis (PCA) were utilised to elucidate multivariate relationships between taxa and determine the extent of taxon differentiation. Therefore, it is of great interest to apply the techniques to support species identification of Southeast Asian *Peperomia*.

## 2. Results

The HPTLC analysis of 14 samples of Peperomia revealed various phytochemicals (Figure 1). The chromatograms from the hexane extracts were obtained after scanning at UV 254 nm (Figure 1a) and UV 366 nm (Figure 1b) and after spraying with anisaldehyde–sulfuric acid reagent (Figure 1c). Different patterns were evident among the 14 species.

The use of multiwavelength imaging of the HPTLC plates, along with the segmentation of images into grey (K), red (R), green (G) and blue (B) channels, enhanced selectivity and provided extensive additional information for distinguishing compounds based on their fluorescent characteristics (Figure 2). This approach is particularly advantageous for complex mixtures containing compounds with varying absorbance properties, as observed in the analysed extracts. Multiwavelength imaging, when combined with different colour scale techniques, was beneficial in revealing the presence of diverse compounds. Colour scale fingerprints (Figure 2), which describe changes in optical density and signal intensity corresponding to the concentration of mixture components, were employed as variables for chemical profile analysis in this investigation. Images acquired under UV wavelengths of 254 nm and 365 nm were utilised for this purpose. Notably, under 254 nm detection, the green channel exhibited higher sensitivity compared to other channels, yielding largely similar profiles across the observed compounds.

The cluster dendrogram of 14 species of *Peperomia* from the hexane extract colour scale fingerprint dataset is shown in Figure 3. The h-clust function in the R programming language was used to create hierarchical clustering, a technique for grouping similar data points together based on distance or similarity measures [23]. In the hierarchical cluster analysis based on HPTLC fingerprinting, we selected a horizontal line at a height of 600. This height represents the similarity of sample groups using the Complete Linkage method, which calculates the distance between groups by considering the farthest points within each group [24,25]. Typically, a partition can be selected at a specific level, where the number of clusters corresponds to the desired value. To achieve this, the dendrogram was cut at the appropriate level [26]. In this study, we primarily focused on a height of 600 for our analysis.

At the distance between the species of 600 (height = 600), it was found that *P. tetraphylla* and *P. ranongensis* were distinguished from the other species. Moreover, *P. laevifolia* and *P. kotana* were clearly separated from other species. *P. pellucida* and *P. sirindhorniana* were also distinguished from the other species in the subcluster. *P. masuthoniana*, *P. cavaleriei*, *P. heyneana* and *P. dindygulensis* were placed close to each other and belong to the same group in the cluster dendrogram, together with *P. moulmeiniana*, *P. nakaharai, P. heptaphylla* and *P. multisurcula*.

The PCA analysis differed from the cluster analysis, with PC1 and PC2 accounting for a total of 66.56% of the variation. The PCA can be divided into four groups: *P. laevifolia* and *P. kotana* were clustered together in one group, *P. tetraphylla* was separated into an independent group, and *P. ranongensis* formed its own independent group. Meanwhile, a larger group comprised 10 species (Figure 3b).

The HPTLC multiwavelength imaging using ethyl acetate extract separated the 14 Peperomia into seven groups, which generally corresponded with those groups obtained with the hexane extract but sometimes contrasted with that analysis (Appendix A). Furthermore, the cluster dendrogram between 14 species of *Peperomia* using the ethanol extract divided the samples into six groups (Appendix A).

The clustering dendrogram based on colour scale HPTLC fingerprinting from the combined data from the three solvent extracts is shown in Figure 4. The 14 species could all be clearly separated at a distance of 600 (height = 600). The PCA analysis using data from three solvents showed that the PC1 and PC2 axes explained 58.63% of the total variance. The PCA successfully separated eight *Peperomia* species, except for three pairs: *P. dindygulensis* and *P. heyneana*, *P. multisurcula* and *P. heptaphylla*, *P. nakaharai* and *P. moulmainiana*, which clustered together, indicating a trend of chemical similarities between them (Figure 4b). However, when combined with the cluster dendrogram, the data can effectively support the differentiation of *Peperomia* species from Southeast Asia.

The phytochemical composition of the hexane extract of 14 species of *Peperomia* native to Southeast Asia was analysed by gas chromatography–mass spectrometry (GC-MS). The GC-MS spectrum data showed several peaks (Appendix A) that indicated the presence of 40 different compounds with retention time ranging from 4.04 to 47.51 min (Table 1). Each of these chemicals was recognised and described using the mass fragmentation patterns and retention indices from the Spectral Library and Database (Wiley 6, NIST02 and NIST17 Library).

The discovered constituents in the *Peperomia* hexane extract have previously been shown to exhibit a variety of advantageous pharmacological and therapeutic activities. The compounds have been shown to have anti-tumour, antibacterial, anti-inflammatory, analgesic, antineoplastic, anti-virulence, antioxidant, antifungal, abortifacient, antiandrogenic and anticancer activities [27]. 

Gamma–Sitosterol was identified as a common compound present in all 14 *Peperomia* species (Table 1). Additionally, compounds such as Linoelaidic acid, 7-Tetradecenal (Z)- and Tetra-tetracontane were found in many *Peperomia* species. Furthermore, certain compounds were unique to specific *Peperomia* species: Humulene and (−)-Spathulenol were exclusively found in *P. tetraphylla*; Carotol and Apiol were specific to *P. pellucida*. Germacrene D was uniquely identified in *P. dindygulensis*; Eicosanoic acid was solely detected in *P. cavaleriei*; and Hexacosanal was found only in *P. multisurcula*. Among the compounds identified, 20 compounds have been previously reported in *Peperomia* species, and 6 compounds have been identified in *Piper* species. Notably, 20 compounds were identified that have not been previously described in *Peperomia* species (Appendix A).

The cluster dendrogram from the GC-MS chemical compound data divided the 14 species into six groups with a distance between the species of 0.6 (height 0.6). *P. masuthoniana* was separated as a non-grouped individual species (group 1). *P. dindygulensis* and *P. heyneana* formed group 2. *P. heptaphylla* and *P. laevifolia* formed group 3. Group 4 consisted of *P. ranongensis*, *P. pellucida*, *P. kotana* and *P. sirindhorniana*. Group 5 consisted of two species, namely *P. moulmeiniana* and *P. nakaharai*. Group 6 included *P. cavaleriei*, *P. multisurcula* and *P. tetraphylla*. Subsequent PCA found that the combined PC1 and PC2 axes were able to explain 36.43% of the total variance. The plot shows that the 14 species can be divided into six main groups that were also found in the HCA (Figure 5).

## 3. Discussion

HPTLC and GC-MS are powerful techniques for separating, identifying and quantifying chemical components in plants, and we have demonstrated their utility in a diverse group of *Peperomia* species. They have also been optimised in other studies for the separation and identification of phenolic compounds, providing a rapid, precise and cost-effective analysis [28]. Furthermore, HPTLC has been applied to the quality control of botanicals, demonstrating its efficiency and sensitivity [29]. On the other hand, GC-MS has been extensively used in the analysis of natural products, including medicinal plants, due to its selectivity, sensitivity and speed. Our studies therefore add to these studies that collectively highlight the reliability and utility of HPTLC and GC-MS in the identification and analysis of plant species.

The chromatograms from the hexane extracts, obtained after scanning at UV 254 nm (Figure 1a) and following treatment with anisaldehyde–sulfuric acid reagent (Figure 1c), exhibited similar trends. *P. laeviflolia* and *P. masuthoniana* displayed clearer and more distinct bands compared to the other species. Under UV 366 nm observation (Figure 1b), *P. masuthoniana* exhibited a red fluorescence band, typically indicative of flavonoids or other aromatic compounds [30]. The results obtained from high-performance thin-layer chromatography (HPTLC) using only ethyl acetate separated the compounds into seven groups, which generally corresponded with those identified in the hexane extract, suggesting a similar trend between low-polarity and medium-polarity compounds. In contrast, the group of more polar compounds extracted with ethanol showed a markedly different separation pattern.

The results obtained from HPTLC using three different solvents, hexane, ethyl acetate and ethanol, revealed distinct chemical profiles and were able to group taxa in the cluster analysis. However, *P. tetraphylla* and *P. ranongensis* did not cluster closely to other taxa with the hexane extract and ethyl acetate extract. In the same way, *P. laevifolia* did not cluster closely with other taxa in the ethyl acetate extract or the ethanol extract. The results indicate that *P. tetraphylla, P. ranongensis* and *P. laevifolia* have distinctly different chemical profiles from another *Peperomia* species.

The chemical profile analysis using GC-MS identified 40 chemical compounds extracted from various *Peperomia* species using hexane. Gamma–Sitosterol was found consistently across all 14 species studied. Two previous studies [31,32] reported the presence of Apicol and Caryophyllene, respectively, in *P. pellucida*, which aligns with our findings of these compounds in the same species. Furthermore, our research found Caryophyllene in *P. cavaleriei*, *P. masuthoniana* and *P. tetraphylla*, which had not been previously reported. In contrast to the findings [31] of 9,12-Octadecadienoic acid (Z,Z)-methyl ester in *P. pellucida*, our study did not detect this compound in *P. pellucida* but found it in *P. moulmeiniana*, *P. leavifolia* and *P. cavaleriei*. Additionally, one report [33] revealed the presence of phytol in *P. pellucida*, which contrasts with our results. Instead, we found phytol in *P. ranongensis* and *P. leavifolia.*

The data from GC-MS were subjected to binary scoring analysis, where the presence or absence of compounds was represented. Subsequently, PCA was conducted and the outcome (PC1 and PC2 explaining 36.43 of the total variance) significantly differs from the PCA conducted from HPTLC analysis of hexane extracted from *Peperomia*, which could explain up to 66.56% of the data variance in PC1 and PC2. These results indicate that different analytical methods afford distinct outcomes in terms of chemical constituents and the extent of data coverage. The higher explanatory data from the colour scale fingerprints by HPTLC suggest that it might provide a more comprehensive understanding of the chemical composition and diversity within the analysed samples. This underscores the importance of selecting appropriate analytical techniques tailored to the specific research objectives and data complexity.

## 4. Materials and Methods

### 4.1. Plant Samples 

The fresh samples and specimens were collected from field sites throughout Thailand for *Peperomia* representative of Southeast Asia [3] (Table 2). Herbarium voucher specimens were prepared and deposited in BK, BKF and QBG (acronyms of herbaria follow Thiers [34] Index Herbariorum). Two hundred grams of leaves and stems of *Peperomia* was cut into small pieces and shade-dried at room temperature. The material was ground to a coarse powder using an electronic grinder. The samples were stored at room temperature at the Department of Botany, Faculty of Science, Kasetsart University, Thailand. 

### 4.2. Extraction 

Analytical-purity-grade hexane, ethyl acetate and ethanol (CT Laboratory, Bangkok, Thailand) were used as solvents for extraction, mobile phase preparation and plate derivatisation. The plant materials were powdered (50 g/each) and macerated in 200 mL of n-hexane for seven days at room temperature. Ethyl acetate and ethanol were also used as alternative solvents for extraction using the same materials and the same method as outlined for hexane. The extracts were filtered through filter paper and concentrated using a rotary evaporator at 38 °C to obtain solid crude extracts. 

### 4.3. High-Performance Thin-Layer Chromatography (HPTLC)

HPTLC was employed for the generation of chemical profiles. TLC was performed with pre-coated silica gel 60 PF254 plates (Merck, Darmstadt, Germany). The crude extracts of leaves and stem were diluted to 10 mL and spotted on TLC plates using an automatic TLC sampler. TLC was performed in a TLC tank and observed under white light and UV (wavelengths 254 and 366 nm). The relative front (Rf) value was then recorded, and photographs were taken at each wavelength. Anisaldehyde–sulfuric acid reagent was used for the detection of terpenoid, steroids, phenols and sugars; positive tests after spraying show violet or blue spots (phenol), red (terpene), grey (sugar) and green (steroid). 

### 4.4. Image Processing

The TLC Analyser Digital Scanning software (Version 1.1) was used for image processing and digitised chromatogram data acquisition (https://www.sciencebuddies.org/science-fair/projects/competitions/tlc-analyzer, accessed on 20 August 2023). This program virtually scans across the obtained JPG image, acting as a simulated TLC scanner that scans the surface of the chromatographic plate along with the developed track. The following parameters were selected for the scanning process: eye dropper size 3 × 3; left margin 350; right margin 250. At each measurement point, the intensity of the reflected light was recorded. Taken together, the measurements formed the densitograms describing changes in the optical density and intensity of the signal along each line. Image density values for pure RGB colour (red, green and blue channel) and for black and white (K, grey channel) were plotted by multispectral scans to provide corresponding colour scale fingerprints. The numerical values obtained by digitisation of the individual RGB and K spectral scans were further used as initial variables in the chemometric analyses. This generated a total of 5263 continuous characters per single species and, overall, it generated 73,656 characters.

### 4.5. Gas Chromatography–Mass Spectrometry (GC-MS)

GC-MS chromatograms were recorded using a Hewlett Packard Model 5975B mass spectrometer (MSD; Agilent Technologies, Santa Clara, CA, USA). The GC was equipped with a ZB-5 column (30 m × 0.53 mm × 1.50 µm). The GC parameters were as follows: the carrier gas was highly pure helium with a 1 mL/min flow rate. The inlet temperature was 250 °C with a split ratio of 20:1 and the pressure was 49.7 kPa. The column oven temperature was initially set at 40 °C for 1 min, ramped to 290 °C at 5 °C/min, and kept at 290 °C for 5 min. MS parameters were as follows: data were acquired in electron impact (EI) mode, using the full scan mode from *m*/*z* 40 to 750. The n-hexane components were identified based on their linear retention indices (RIs), determined experimentally relative to the tR of n-alkanes (C8–C40) on the same column and the NIST Standard Reference Database 69: NIST Chemistry WebBook. Mass spectra data were compared against commercially available MS libraries Wiley 6, NIST02 and NIST 17. 

### 4.6. Statistical Analyses 

The results from differing extracts (hexane, ethyl acetate and ethanol extracts) of 14 samples of *Peperomia* from Thailand were analysed by principal component analysis (PCA) and hierarchical clustering analysis (HCA). The PCA was carried out using R code to compute and visualise the statistics using the prcomp() function and the facto extra package in posit cloud [35]. The HCA was generated using the hclust method in R, which measures the distance between clusters based on the sum of the squared distances between all pairs of data points in the clusters. The height of the dendrogram represents the distance between the clusters.

## 5. Conclusions

This study demonstrates that traditional morphological methods alone are insufficient for distinguishing between indigenous Southeast Asian *Peperomia* species due to their similar floral structures, differences between fresh and dried material, and the fact that reproductive organs are often lost when specimens are dried. However, the integration of chemometric data, specifically through HPTLC fingerprinting and GC-MS analysis, offers a robust solution for species identification and classification. The use of a combined solvent dataset along with PCA and HCA has proven effective in differentiating the *Peperomia* taxa. The GC-MS analysis further revealed 40 distinct compounds that varied among species. Twenty compounds were identified that have not been previously described in *Peperomia* species. The GC-MS technique was not able to distinguish species levels. Overall, this multifaceted analytical approach significantly improves the reliability of *Peperomia* species authentication and can be effectively applied in future research and conservation efforts.

## Figures and Tables

**Figure 1 plants-13-02751-f001:**
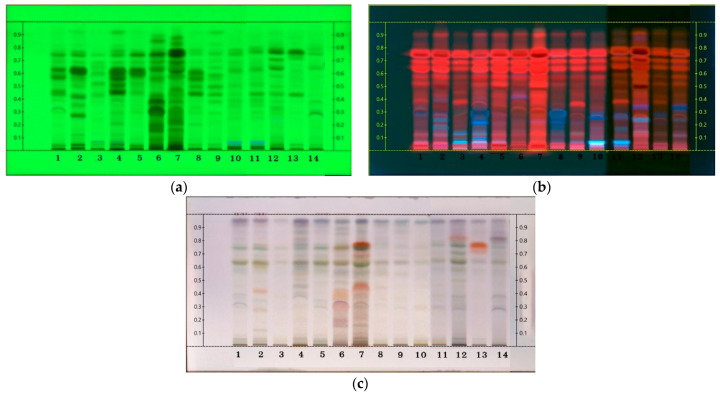
HPTLC profile of Peperomia from hexane extracts; (**a**) scanned at UV 254 nm; (**b**) scanned at UV 366 nm; and (**c**) scanned after spraying with anisaldehyde–sulfuric reagent. The *Y*-axis shows the Rf values for each compound and the *X*-axis shows the species code: (1) *P. cavaleriei*; (2) *P. dindygulensis*; (3) *P. heptaphylla*; (4) *P. heyneana*; (5) *P. kotana*; (6) *P. laevifolia*; (7) *P. masuthoniana*; (8) *P. moulmeiniana*; (9) *P. multisurcula*; (10) *P. nakaharai*; (11) *P. pellucida*; (12) *P. sirindhorniana*; (13) *P. tetraphylla*; (14) *P. ranongensis*.

**Figure 2 plants-13-02751-f002:**
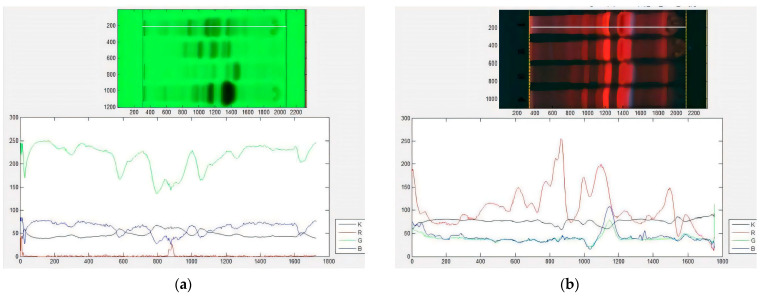
Image of chromatographic plates and corresponding colour scale fingerprints for Peperomia cavaleriei. Fingerprints were generated using TLC Analyzer software and different colour scale selections in (**a**) fluorescence quenching mode under 254 nm excitation wavelength and (**b**) fluorescence quenching mode under 366 nm excitation wavelength. Red line—red scale fingerprint (R); green line—green scale fingerprint (G); blue line—blue scale fingerprint (B); grey line—grey scale fingerprint (K).

**Figure 3 plants-13-02751-f003:**
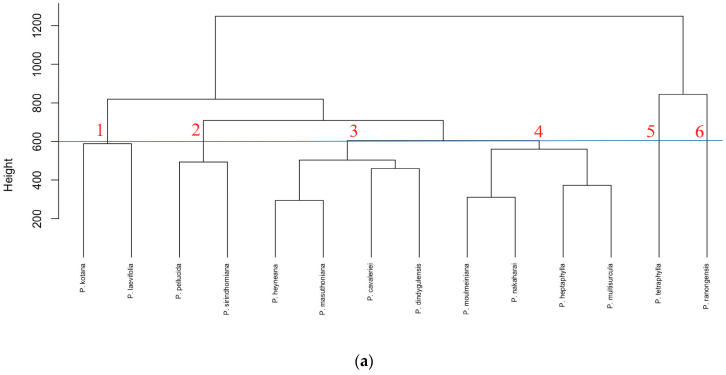
Clustering dendrogram and PCA of the HPTLC results based on the colour scale fingerprint data using the hexane extracts for 14 Peperomia samples; (**a**) clustering dendrogram (numbers 1 to 6 is cluster group); (**b**) PCA profile (PC1 accounts for 56% of the variation; PC2 accounts for 10.5% of the variation).

**Figure 4 plants-13-02751-f004:**
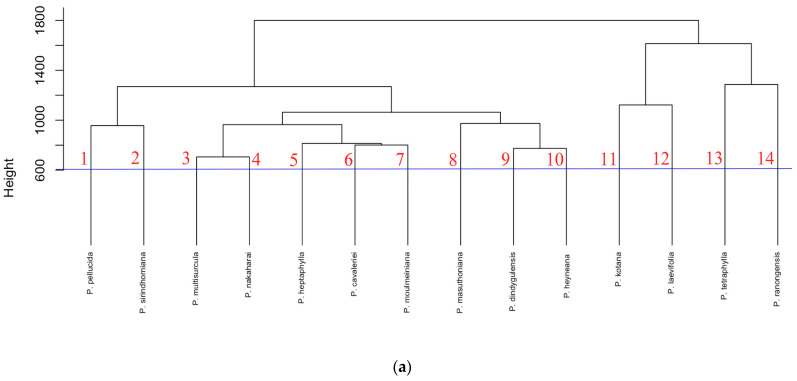
Clustering dendrogram (**a**) and PCA profile (**b**) of the HPTLC based on colour scale fingerprint data using the combined data from the three different solvent extracts of 14 Peperomia samples.

**Figure 5 plants-13-02751-f005:**
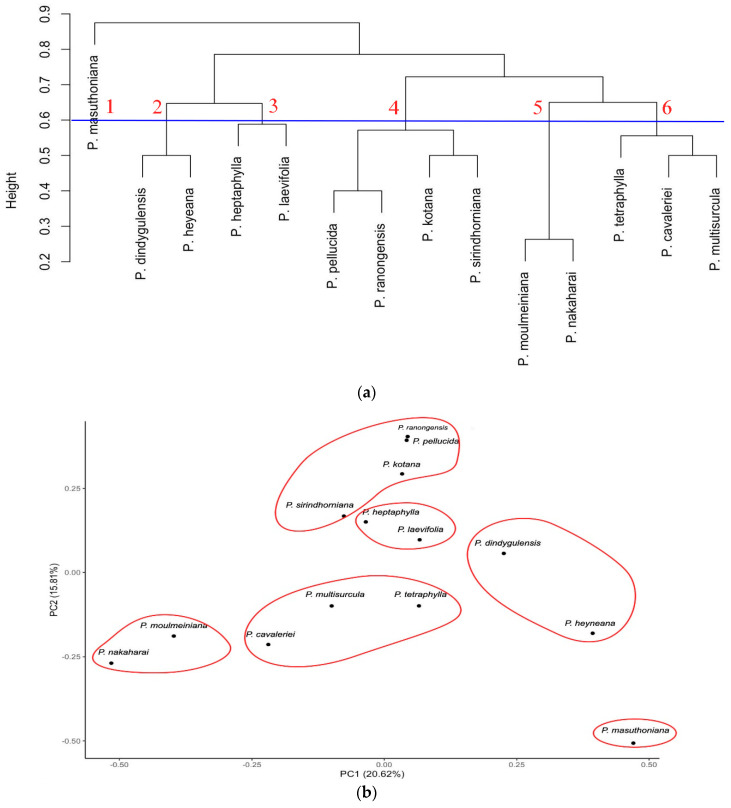
Clustering dendrogram (**a**) and principal component analysis (**b**) ordination of the chemical composition data from the 14 n-hexane extract of *Peperomia* samples assessed by GC-MS. Axis PC1 accounts for 20.62% and axis PC2 for a further 15.81% of the total variance.

**Table 1 plants-13-02751-t001:** Composition of the n-hexane extract of *Peperomia* from Southeast Asia assessed by the GC-MS data.

Compound	Retention Time	Peak Area %	Species Code Found	Identification
alpha.-Pinene	4.04	1.02	4, 7	RI, MS
beta.-Pinene	4.74	1.77–1.92	2, 4, 6, 7	RI, MS
Copaene	14.13	1.79–2.54	3, 4, 14	RI, MS
Caryophyllene	15.24	2.21–3.55	1, 7, 11, 13	RI, MS
Humulene	16.09	8.03	13	RI, MS
Germacrene D	16.74	8.24	2	RI, MS
Pentadecane	17.02	1.3	11	RI, MS
Beta-Bisabolene	17.31	2.56	1	RI, MS
(-)-Spathulenol	19.02	1.74–2.11	13	RI, MS
Carotol	19.48	3.75	11	RI, MS
8-Heptadecene *	21.16	8.77–17.3	8, 10	RI, MS
Apiol	21.29	1.47	11	RI, MS
Eicosane *	21.60	3.88–3.95	8, 10	RI, MS
n-Hexadecanoic acid	26.99	4.59–8.31	1, 3–13	RI, MS
Hexadecanoic acid, ethyl ester	27.60	3.24–5.77	1, 3, 6, 8, 9, 10, 12	RI, MS
Phytol	29.82	2.28–3.15	6, 14	RI, MS
Linoelaidic acid *	30.13	1.29–6.90	1–3, 5–12, 14	RI, MS
7-Tetradecenal, (Z)- *	30.25	4.56–5.84	2–6, 8–11, 13, 14	RI, MS
cis,cis,cis-7,10,13-Hexadecatrienal *	30.29	6.11–7.33	1, 9, 10	RI, MS
Linoleic acid ethyl ester *	30.68	4.20–4.65	3, 10	RI, MS
9,12,15-Octadecatrienoic acid, (Z,Z,Z)-	30.79	2.99–4.31	1, 3	RI, MS
9,12-Octadecadienoic acid (Z,Z)-	30.96	1.55–6.08	1, 6, 8	RI, MS
Eicosanoic acid *	31.22	1.43	1	RI, MS
Octadecanoic acid, ethyl ester *	31.25	1.29–2.10	2, 3, 9, 11, 14	RI, MS
(E)-1-(2-Hydroxy-4,6-dimethoxyphenyl)-3-phenylprop-2-en-1-one *	38.37	3.38–3.79	8–10	RI, MS
Tetracosane *	39.25	3.04–8.16	1, 7, 9, 10, 13	RI, MS
Flavone, 5-hydroxy-7,8-dimethoxy- *	39.69	4.51–4.54	8, 10	RI, MS
4H-1-Benzopyran-4-one,5,7-dimethoxy-2-phenyl- *	40.60	1.98–2.15	8, 10	RI, MS
Hexacosanal *	41.18	1.81–2.71	9	RI, MS
Epicholestanol *	41.24	8.78	5	RI, MS
Tetratetracontane *	42.03	4.17–12.51	1–4, 6–14	RI, MS
Octacosanal *	43.89	1.47–6.87	1–4, 7–10, 13	RI, MS
Hentriacontane *	44.64	3.33–8.11	1–2, 4, 6, 8–14	RI, MS
1-nonanone, 1-(2,5-dihydroxyphenyl)- *	45.29	1.92–10.55	2, 4, 7	RI, MS
(+)-Sesamin *	45.46	2.08–9.63	7, 9	RI, MS
Campesterol	46.41	3.12–3.60	3, 6	RI, MS
Ergost-5-en-3-ol, (3.beta.)- *	46.43	1.71–6.67	1, 5, 10–14	RI, MS
E,E-2,13-Octadecadien-1-ol	46.45	5.07	7	RI, MS
Stigmasterol	46.80	3.1–3.77	1–3, 5, 8–14	RI, MS
gamma.-Sitosterol	47.51	2.74–8.02	1–14	RI, MS

Note: * means compounds that have solely been identified in the extract of the *Peperomia* in this study. Species code: (1) *P. cavaleriei*; (2) *P. dindygulensis*; (3) *P. heptaphylla*; (4) *P. heyneana*; (5) *P. kotana*; (6) *P. laevifolia*; (7) *P. masuthoniana*; (8) *P. moulmeiniana*; (9) *P. multisurcula*; (10) *P. nakaharai*; (11) *P. pellucida*; (12) *P. sirindhorniana*; (13) *P. tetraphylla*; (14) *P. ranongensis*.

**Table 2 plants-13-02751-t002:** List of *Peperomia* species used for chemometric investigation.

No.	Species	Province	Collector No.
1	*P. cavaleriei*	Tak	*Y. Banchong 174* (BK, BKF, QBG)
2	*P. dindygulensis*	Phayao	*Y. Banchong 151* (BK, BKF, QBG)
3	*P. heptaphylla*	Prachuap Khiri Khan	*Y. Banchong 136* (BK, BKF, QBG)
4	*P. heyneana*	Chiangmai	*Y. Banchong 139* (BK, BKF, QBG)
5	*P. kotana*	Nakhon Si Thammarat	*Y. Banchong 157* (BK, BKF, QBG)
6	*P. laevifolia*	Nakhon Si Thammarat	*Y. Banchong 159* (BK, BKF, QBG)
7	*P. masuthoniana*	Chiangmai	*Y. Banchong 143* (BK, BKF, QBG)
8	*P. moulmeiniana*	Kanchanaburi	*Y. Banchong 161* (BK, BKF, QBG)
9	*P. multisurcula*	Nan	*Y. Banchong 148* (BK, BKF, QBG)
10	*P. nakaharai*	Chiangmai	*Y. Banchong 169* (BK, BKF, QBG)
11	*P. pellucida*	Bangkok	*Y. Banchong 170* (BK, BKF, QBG)
12	*P. sirindhorniana*	Loei	*Y. Banchong 152* (BK, BKF, QBG)
13	*P. tetraphylla*	Chiangmai	*Y. Banchong 135* (BK, BKF, QBG)
14	*P. ranongensis*	Ranong	*Y. Banchong 63* (BK, BKF, QBG)

## Data Availability

The original contributions presented in the study are included in the article/Appendix A, further inquiries can be directed to the corresponding author.

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
