# Peer review of "Chemotaxonomy of Southeast Asian Peperomia (Piperaceae) Using High-Performance Thin-Layer Chromatography Colour Scale Fingerprint Imaging and Gas Chromatography–Mass Spectrometry"

_plants, 2024, doi:10.3390/plants13192751_

Round 1

Reviewer 1 Report

Comments and Suggestions for Authors

See attachment.

Author Response

Comment 1 :  Keywords: I suggest keeping chemometrics as a keyword without approaches and also adding HPTLC and GC-MS as keywords.

Response 1: We agree with this comment and add to the revised manuscript in Keyword line 29.

Comment 2 :  L33-40. Please provide references for these statements

Response 2: We agree with this comment and add reference [2] and [3]on that phase at line 33-40

Comment 3 : The introduction is so short that it fails to provide state-of-the-art analytical methods, chemometric tools, and plant profiling within this field.

Response 3:  We agree with this comment and add the text in line 46-50, 72-73

Comment 4: L108. The use of

Response 4: We agree with this comment and change to the revised manuscript in line 107.

Comment 5: Figure 2. Please enhance the resolution of the chromatograms.

Response 5: We agree with this comment and change the figure that edit the resolution of the chromatograms.

6) Comment 6: Figure 3. I want to ask the author under which criteria they segregate the P. dindygulensis and P. heptaphylla samples in the PCA chart; according to the provided plot, they have pretty similar answers in all responses; thus, I suggest either clustering them together or keeping them outside the corresponding clusters.

Response 6:  The grouping in PCA was based on the dendrogram, where it is evident that P. dindygulensis and P. heptaphylla are on separate nodes when considering a height of 600. To maintain consistency across all test results, we applied this height criterion uniformly. As shown in Figures 3 and 5, the results are consistent, and therefore, we decided to follow the clustering pattern as determined by the cluster dendrogram.

Comment :7  Figure 4. Similarly, how did the authors decide that none of the samples is clusterable? It seems that P. Dindygulensis and P. heyneana may be forming a cluster, as well as P. multisurcula and P. heptaphylia. Please revise this.

Response 7: There is a tendency for P. dindygulensis and P. heyneana to be grouped together when considering heights greater than 600, similar to the grouping of P. multisurcula and P. heptaphylla. However, since we are committed to analyzing the data at a height of 600, P. dindygulensis and P. heyneana, as well as P. multisurcula and P. heptaphylla, appear as separate nodes according to the cluster dendrogram analysis.

Comment 8: Section 4.1. please specify the number of samples and the storage conditions before, during, and after the analysis.

Response 8: We agree with this comment and change to the revised manuscript at line 386-387.

Comment 9: Section 4.2. Although maceration is a well-established extraction method, please reference the extraction conditions. Also, I want to ask the authors why they chose this time-consuming, solvent-consuming method when faster and more environmental extraction methods are available.

Response 9: The maceration process was conducted using the closed glass chamber, room temperature and maceration for 7 days. These conditions were selected based on their proven effectiveness in extracting the target compounds. Despite being time and solvent-intensive, maceration was chosen for its ability to extract a wide range of compounds comprehensively. It also ensures the preservation of thermolabile constituents and provides a consistent basis for comparison with existing literature. Future studies may explore faster, more eco-friendly methods once the full compound profile is established.

Comment 10: Section 4.5. Please fully describe the instrumental conditions, including injector temperature, type of liner, injection mode, and volume, split ratio (if applicable), temperature program, carried gas flow, transfer line temperature, type and temperature of the source, ionization mode, and type and temperature of the quadrupole.

Response 10 : We addition the The GC parameters were as follows to Section 4.5: the carrier gas was highly pure helium with a 1 mL/min flow rate. The inlet temperature was 250°C with a split ratio of 20:1, and the pressure was 49.7 kPa. The column oven temperature was initially set at 40°C for 1 min, ramped to 290°C at 5°C/min, and kept at 290°C for 5 min. MS parameters were as follows: data were acquired in the electron impact (EI) mode, using the full scan mode from m/z 40 to 750.

Comment 11: Please add a GC-MS chromatogram of each plant sample.

Response 11: We agree with this comment and addition in the supplementary file as Figure S3-S16 .

Comment 12: The conclusion section is poor; what are the novelties of this work? What would its implications be? What research questions were answered

Response 12: We agree with this comment and revise the conclusion in line 439-453.

Comment 13: The list of references is short and outdated; this is problematic because it may lead to a lack of contextualization, making it difficult to identify the real novelties of this work. Please update references 1-4, 12-15, 18-19,23 to not older than 2014. More references are expected after enhancing the introduction part of this work as required in this revision, which should match the same criteria.

Response 13: We would like to clarify that the references used in our work are appropriate and reliable sources, selected from reputable databases and thoroughly reviewed. Although some references are older than 2014, the information presented in our paper is still relevant and important for our study. Since the plants we studied are relatively few in scientific literature, the currently available information remains an important source for citing and analyzing in this research.

Reviewer 2 Report

Comments and Suggestions for Authors

The authors of the current manuscript tackled the intriguing and contemporary subject of identifying the fingerprints of selected species from the genus Peperomia, some of which find application in traditional medicine. To define characteristic marker compounds, they tested different extracts and applied HPTLC results for hierarchical and principal component analysis. Additionally, they run hexane extracts on GCMS and analyse them in detail.

In the introduction section, the authors provided a brief summary of the current advancements in systematics of this genus, omitting specific details provided by other authors and focusing on the primary problem that guided their work.

Authors should give more information about the importance of these species for the pharmaceutical industry and/or traditional medicine. The authors should present a critical analysis of studies that have examined the chemical composition of the same species from various localities, as well as the variations in metabolism resulting from different phenophases. This is crucial because it represents a critical review of the previous results and gives weightiness to their study.

The majority of the result section is well presented with informative Figures. The HPTLC subsection was systematic and easy to understand; the newer the less multivariate analysis needs to be refined. The results of PCA analysis should be more detailed, with how many principal components the entire data set was described and in what percentage. It is necessary to highlight the grouping within the PCA model, then compare those results with the cluster analysis. The same comment applies to the results of the multivariate analysis of GC/MS data. In cluster analysis, the authors mention height values without commenting on them.

In case comparative analyses of samples of the same species from different localities have not been done so far, this should have been done in this study. In addition, the authors should have stated why and how they decided on these species (specimens), whether they have any special significance, whether they represent a problem for systematists, etc.

Finally, the big question is why the authors distinguished themselves by collecting and preserving plant material in this way. It is recommended that in such research the procedure of suddenly stopping the metabolism of the harvested bull with liquid nitrogen or, if this is not possible, by drying with silica gel. Uneven weather conditions due to harvesting can lead to prolonged effects on enzyme activities.

Considering that in the HPTLC plates and cluster analysis, the authors numbered individual samples (species) with numbers from 1 to 14, it would be good to rename the penultimate column in table 1 (such as the species where it was detected) and enter the numbers corresponding to that species. In cases where a compound is valid in all types, do not list them but indicate 1-14, etc.

The text below Figure 5. does not indicate which part of the picture it refers to.

The authors used three solvents of different polarity, so it was expected that they would not get the same cluster analysis results because they did not extract the same compounds. Only hexane extracts are included in GCMS analysis; why? If, with adequate preparation/derivatisation, the analysis of the other extracts had been carried out, much more information would have been obtained and the results of the HPTLC analysis could be explained.

Precisely all the mentioned shortcomings in the setup and execution of the experiments caused the inability to reach a concrete and valid conclusion.

Author Response

Comment 1: Authors should give more information about the importance of these species for the pharmaceutical industry and/or traditional medicine. The authors should present a critical analysis of studies that have examined the chemical composition of the same species from various localities, as well as the variations in metabolism resulting from different phenophases. This is crucial because it represents a critical review of the previous results and gives weightiness to their study.

Response 1: We agree with this comment and addition the information about the importance of these species for the pharmaceutical industry and/or traditional medicine in the line 46-49. For the collection on various locality and difference phenophases in this genus has several limitations because many Peperomia species are found exclusively in specific areas and are not present in other regions. As a result, samples could only be collected from a single site. Additionally, this study focuses solely on explaining the differences between species.

Comment 2: The authors should present a critical analysis of studies that have examined the chemical composition of the same species from various localities, as well as the variations in metabolism resulting from different phenophases. This is crucial because it represents a critical review of the previous results and gives weightiness to their study.

Response 2 This research does not focus on the variability of sample collection sites in relation to the key compounds. Instead, it prioritizes the use of HPTLC fingerprinting and GC-MS to explain differences at the species level. While the point regarding site variability is interesting, it will be addressed in subsequent studies.

Comment 3: The majority of the result section is well presented with informative Figures. The HPTLC subsection was systematic and easy to understand; the newer the less multivariate analysis needs to be refined. The results of PCA analysis should be more detailed, with how many principal components the entire data set was described and in what percentage. It is necessary to highlight the grouping within the PCA model, then compare those results with the cluster analysis. The same comment applies to the results of the multivariate analysis of GC/MS data. In cluster analysis, the authors mention height values without commenting on them.

Response 3: The h-clust function in the R programming language is used for creating hierarchical clustering, a technique for grouping similar data points together based on distance or similarity measures. In the hierarchical cluster analysis based on HPTLC fingerprinting, we selected a horizontal line at a height of 600. This height represents the similarity of sample groups using the Complete Linkage method, which calculates the distance between groups by considering the farthest points within each group. This approach effectively demonstrates the clear differentiation of Peperomia groups. Typically, a partition can be selected at a specific level, where the number of clusters corresponds to the desired value. To achieve this, the dendrogram is cut at the appropriate level. In this study, we primarily focused on a height of 600 for our analysis.

Comment 4: In case comparative analyses of samples of the same species from different localities have not been done so far, this should have been done in this study. In addition, the authors should have stated why and how they decided on these species (specimens), whether they have any special significance, whether they represent a problem for systematists, etc.

Response 4: In this study, we were unable to collect samples of the same species from multiple locations because many Peperomia species are found exclusively in specific areas and are not present in other regions. As a result, samples could only be collected from a single site. Additionally, this study focuses solely on explaining the differences between species.

Comment 5: the big question is why the authors distinguished themselves by collecting and preserving plant material in this way. It is recommended that in such research the procedure of suddenly stopping the metabolism of the harvested bull with liquid nitrogen or, if this is not possible, by drying with silica gel. Uneven weather conditions due to harvesting can lead to prolonged effects on enzyme activities.

Response 5 : Given the large quantity of samples we collected, using silica gel presented certain limitations. Air-drying in the shade allowed the samples to dry gradually. This study aims to identify the key compounds present in the entire plant, and drying under these conditions ensures that these compounds are not degraded by heat, as would occur with high-temperature oven drying or direct sunlight exposure. Concerns about enzymatic alteration were not a priority, as traditional medicine has long employed this method for drying herbs without adverse effects.

Comment 6: Considering that in the HPTLC plates and cluster analysis, the authors numbered individual samples (species) with numbers from 1 to 14, it would be good to rename the penultimate column in table 1 (such as the species where it was detected) and enter the numbers corresponding to that species. In cases where a compound is valid in all types, do not list them but indicate 1-14, etc.

Response 6 : We agree with this comment and arrange the numbered of individual samples and showed that in the column of table 1.

Comment 7: The text below Figure 5. does not indicate which part of the picture it refers to.

Response 7 : We agree with this comment and add the texted (a.) and (b.)

Comment 8: The authors used three solvents of different polarity, so it was expected that they would not get the same cluster analysis results because they did not extract the same compounds. Only hexane extracts are included in GCMS analysis; why? If, with adequate preparation/derivatisation, the analysis of the other extracts had been carried out, much more information would have been obtained and the results of the HPTLC analysis could be explained.

Response 8: Since we aimed to extract all the compounds present in the plant, we chose to use solvents ranging from non-polar to polar in a sequential manner. GC-MS analysis was not performed on the ethyl acetate and ethanol extracts because GC-MS is most effective for analyzing non-polar or slightly polar compounds, such as essential oils and straight-chain hydrocarbons. The compounds in the ethyl acetate and ethanol extracts would be less detectable using this method. Additionally, budget constraints limited us from analyzing all three solvent extracts.

Reviewer 3 Report

Comments and Suggestions for Authors

This research article reports the Chemotaxonomy of Southeast Asian Peperomia using HPTLC color scale fingerprint imaging and GC-MS.

This article can be useful as a tool for authentication and identification studies of Peperomia species to guide further investigations. However, I have some comments about the manuscript.

 Scientific names must be in italics. Lines 104-107, 143-148, 151-152

Please discuss the differences observed for each plant in the chromatograms from the hexane extracts (Figure 1).

Please add to the discussion the results obtained from the extracts with ethyl acetate and ethanol.

 Use Kovats-type gas chromatographic retention indices to ensure the identification of compounds, mainly compounds that have only been identified in the Peperomia extract in this study.

Author Response

Comment 1

Scientific names must be in italics. Lines 104-107, 143-148, 151-152

Response 1 : We agree with this comment and change the scientific names to italic.

Comment 2: Please discuss the differences observed for each plant in the chromatograms from the hexane extracts (Figure 1).

Response2 : We agree with this comment and addition the discussion in line 351-356

Comment 3: Please add to the discussion the results obtained from the extracts with ethyl acetate and ethanol.

Response 3: We agree with this comment and addition the discussion in line 356-361

Comment 4 : Use Kovats-type gas chromatographic retention indices to ensure the identification of compounds, mainly compounds that have only been identified in the Peperomia extract in this study.

Response 4: We utilized Mass Spectrometry and Retention Index to confirm the identity of the compound detected in this study by comparing it with a reference library (Wiley 6, NIST02 and NIST 17) using the GC-MS post-run analysis program.

Round 2

Reviewer 1 Report

Comments and Suggestions for Authors

The authors have addressed almost all issues in the first raw of the revisions, significantly improving their manuscript. Nevertheless, there are still two major issues that must be solved before recommending its publication:

(1) There is a major need for reference updating (see comment on the previous version). Also, provide the DOI for each article in the reference section.

(2) Regarding the interpretation of the PCA, although HCA (dendrogram) is often used in tandem to give a complete picture of the underlying clustering patterns, the interpretation of PCA and HCA must be performed separately and then compared to further explain the findings in the set of data. Thus, my advice is to keep out of the clusters those samples that are not closed enough (or closer between each other), for instance, P. didygulensis and P. heptaphylla samples from Figure 3's PCA, and P. Dindygulensis and P. heyneana from Figure 4's PCA. (observe comments on previous revision). 

Author Response

Comments 1 :  There is a major need for reference updating (see comment on the previous version). Also, provide the DOI for each article in the reference section.

Response 1: We agree with the comments provided. We were unable to find updated references regarding the taxonomy studies of Peperomia (references 4-6) as there are no researchers specifically focused on this genus in the Philippines, Malaysia, or Indonesia, resulting in a lack of recent publications to cite.  As for statements from the FDA, WHO, and EMA, there have been no recent updates beyond the ones cited (References 18-20). We have updated reference 21 to a 2022 source and added reference 17 on applying HPTLC multiwavelength imaging and color scale fingerprinting to reflect more recent trends in the field. Additionally, all references with a DOI have been linked via CrossRef and PubMed.

Comments 2 :  Regarding the interpretation of the PCA, although HCA (dendrogram) is often used in tandem to give a complete picture of the underlying clustering patterns, the interpretation of PCA and HCA must be performed separately and then compared to further explain the findings in the set of data. Thus, my advice is to keep out of the clusters those samples that are not closed enough (or closer between each other), for instance, P. didygulensis and P. heptaphylla samples from Figure 3's PCA, and P. Dindygulensis and P. heyneana from Figure 4's PCA. (observe comments on previous revision).

Response 2 : We acknowledge the comment and agree that the clustering from cluster analysis does not have to correspond directly with the PCA results. Therefore, we have re-explained the grouping in the PCA. In Figure 3, the PCA can be divided into four groups: P. laevifolia and P. kotana are clustered together in one group, P. tetraphylla forms an independent group, and P. ranongensis also forms its own distinct group. Meanwhile, a larger group comprises ten species. In Figure 4, the PCA successfully separated eight Peperomia species, except for three pairs: P. dindygulensis and P. heyneana, P. multisurcula and P. heptaphylla, and P. nakaharai and P. moulmainiana, which clustered together. The above revision has been incorporated into the manuscript, and highlighted in yellow for your reference.

Reviewer 2 Report

Comments and Suggestions for Authors

In my opinion,

The authors significantly improved the manuscript and provided additional evidence through literature references and further explanations. In light of this, I recommend that the editor accept the manuscript i present form.

Author Response

Thank you for your thorough review and positive feedback. We appreciate your acknowledgment of the improvements made to the manuscript. The additional references and explanations provided were aimed at enhancing the clarity and robustness of the paper. We are pleased to hear that the revisions meet your expectations, and we look forward to the editor's decision.

Round 3

Reviewer 1 Report

Comments and Suggestions for Authors

The authors have addressed all issues that arose in the previous revisions; thus, I'm happy to recommend this manuscript for publication in the Plants journal.